# Differential Effect of Four-Week Feeding of Different Dietary Fats on the Accumulation of Fat and the Cholesterol and Triglyceride Contents in the Different Fat Depots

**DOI:** 10.3390/nu12113241

**Published:** 2020-10-23

**Authors:** Min Liu, David Q.-H. Wang, Dennis D Black, Patrick Tso

**Affiliations:** 1Department of Pathology and Laboratory Medicine, University of Cincinnati College of Medicine, Cincinnati, OH 45237, USA; lium@ucmail.uc.edu; 2Department of Medicine and Genetics, Division of Gastroenterology and Liver Diseases, Marion Bessin Liver Research Center, Einstein-Mount Sinai Diabetes Research Center, Albert Einstein College of Medicine, Bronx, NY 10461, USA; david.wang@einsteinmed.org; 3Children’s Foundation Research Institute, Le Bonheur Children’s Hospital, Department of Pediatrics, University of Tennessee Health Science Center, Memphis, TN 38163, USA; dblack@uthsc.edu

**Keywords:** cholesterol, triglyceride, high-fat diet, dietary lipid composition

## Abstract

The aim of the present study was to determine the effects of feeding of a high-fat diet containing different types of lipids for four weeks on the cholesterol and triglyceride contents of different fat depots and on body temperature in rats. Four groups of adult rats were fed 10% fat, containing either beef tallow, safflower oil, or fish oil, respectively, as well as a normal rodent diet with 4% fat, for four weeks. The rats on normal rodent diet consumed significantly more food and water than the rats in the other three groups. Rectal temperature increased only after four-week feeding with safflower oil fat. Increased fat deposition and adipocyte size were observed in rats fed safflower oil and beef tallow. In all fat pads of safflower oil-fed rats, cholesterol content was significantly higher than the other three groups. Feeding of beef tallow increased triglyceride depot without increasing cholesterol content. The rats fed fish oil had significantly less triglyceride and cholesterol deposition in adipose tissues than the rats fed safflower oil or beef tallow. These results clearly demonstrated the differences in fat deposition, adipocyte size and number, triglyceride and cholesterol accumulation in fat cells are dependent on the dietary lipid composition.

## 1. Introduction

Obesity is a growing global health concern that underlies the development of type 2 diabetes, hypertension and cardiovascular disease [1]. Many studies have been performed to define the role of dietary fats on the development of obesity. Compelling evidence from animal studies has demonstrated that chronic feeding with a high-fat diet causes an expansion of adipose tissue mass through an increase in adipocyte size (hypertrophy) and/or number (hyperplasia), leading to the subsequent development of obesity [2]. However, these is little definitive information on the effect of dietary fat type on adipose tissue cellularity and the development of obesity. Su et al. [3] and Hill et al. [4] reported a reduction in fat accumulation in rats fed fish oil, but no differences in energy accumulation between rats fed other dietary fat types, e.g., olive oil vs. beef tallow; or lard vs. corn oil. These observations emphasized the important role of dietary fats with different fatty acid composition on body fat accumulation.

In addition to triacylglycerol, adipose tissue also contains an appreciable amount of cholesterol stored in the esterified form [5,6]. Cholesterol synthesized in the liver is transported to the peripheral tissue by very low-density lipoproteins (VLDL) [7]. Changes in dietary lipid composition affect both high-density lipoprotein (HDL) binding and uptake of cholesteryl ester by adipocytes [8]. Dietary fish oil has the lowering effects on serum cholesterol level [9,10], and clinical studies have confirmed the benefit for cardiovascular diseases [11,12].

However, it is poorly understood whether there is any difference in cholesterol and/or triacylglycerol accumulation in adipocytes of different fat pads. Specifically, it remains unclear about the effects of the chronic feeding of polyunsaturated fatty acids and dietary fish oil on the cholesterol and triacylglycerol content in the fat depots in the subcutaneous and abdominal fat pads. The aim of the present study was to determine the effects of 4-week feeding of a normal rodent diet (4% fat) and high-fat diets, including 10% safflower oil, 10% fish oil or 10% beef tallow, on lipid accretion, including the cholesterol and triacylglycerol content in adipocytes, and on adipose cellularity, including the size and number of the adipocytes, in subcutaneous, epididymal, retroperitoneal and mesenteric fat pads of Sprague-Dawley rats. The effects of normal rodent and different high-fat diets (safflower oil, beef tallow, and fish oil) on food and water intake, as well as body temperature, were also examined.

## 2. Materials and Methods

### 2.1. Animals

Male Sprague-Dawley (SD) rats (180~200 g) were used in this study. Rats were housed in wire-bottomed cages in a room illuminated from 0600 h to 1800 h (hour), 12-h light-dark cycle and maintained at 21 ± 1 °C. All animal procedures were approved by the Institutional Animal Care and Use Committee of the University of Cincinnati.

### 2.2. Experimental Procedures

Three pelleted semi-purified rodent diets (AIN-76) containing 10 g of fat/100 g of diet of either beef tallow, safflower oil, or fish oil, respectively, were prepared at Dyets (Bethlehem, PA). The AIN-76 diet is recommended by the American Institute of Nutrition (AIN) in 1976 as a standard purified rodent diet to study all aspects of health and disease. This type of semi-purified diet is formulated with a more refined and consistent set of ingredients. The nutrient concentrations in the semi-purified diet are less variable and more easily controlled via formulation than in a grain-based diet. A normal rodent diet with 4% fat (soybean oil as primary fat source) from Harlan Labs (Teklad Sterilizable Rodent Diet) was used as control. Twenty-four rats were randomly divided into 4 groups and then fed those 4 different diets for 4 weeks. During the feeding period, food and water intake, body weight and rectal temperature were measured at 11:00 AM.

After 4 weeks of feeding, rats were fasted for 24 h and then anesthetized by halothane. Four fat depots, including epididymal, retroperitoneal, mesenteric and subcutaneous fat, were dissected and collected. Specifically, the epididymal and retroperitoneal fat pads were collected from the right flank. The right epididymal fat pad was removed just distal to the major blood vessel in the base of the pad, and the right retroperitoneal fat pad was removed as a triangular section extending from a vertex in the inguinal region up the midline and across at the lower pole of the kidney, extending laterally as far as fat was visible, according to the previous study reported by Johnson et al. [13]. The subcutaneous fat pad was collected from dorsal to the scapular region in the following manner according to the method reported by Johnson et al. [13]. The lateral cuts were made through the skin from the top of the haunch to the base of the ear, to reveal the underlying fatty layer. At a point about midway of this cut, where the fatty sheath thins markedly, lateral cuts were made into the fat pad following the line of the previous skin incision. Next, a cut was made through the skin below the rib cage across the dorsal surface joining the two lateral incisions. The rectangular flap of skin was then carefully dissected away from the underlying muscle and fascia. The white fat pad was carefully dissected away from the rectangular flap of skin and brown adipose tissue [13].

The dissected white fat pads were washed in saline, dried by Kimwipes and the weight of fat pad was measured and part of each fat pad, approximately 50 mg of wet weight, was removed for evaluation of fat cell size and number by morphology. Remaining fat pads were placed in chloroform-methanol 2:1 for lipid extraction by the method of Folch et al. [14]. Cholesterol and triacylglycerol content of the extracted lipid from each fat pad was determined by chemical assay with Infinity cholesterol (Thermo-Fisher Scientific, Middletown) and TG (Randox TR210) assay kits, respectively, according to manufacturer’s protocols, and as we described previously [15,16].

The small samples from different fat pads were fixed at 37 °C in 25 mL of 0.05 M collidine-HCl buffer (pH 7.4), containing 2 g of osmium tetroxide/l00 mL. Five-days later, the osmium-fixed cells were then washed, collected, and the cell size was determined in isotonic saline with a Coulter electronic counter (Model B, Coulter Electronic, Inc., Hialeah, FL, USA), a commonly used method [17], whereby the size of osmium tetroxide-fixed cells is determined by measurement of electrical resistance. The total number of cells in the sample was calculated by dividing the lipid weight of the fixed tissue sample by the mean adipocyte cell size, as described in method III by Hirsch et al. [18] and modified by Cushman [19].

### 2.3. Statistics

The data are presented as mean ± standard error (S.E.). Data at each time point and from each fat pad were analyzed with GraphPad Prism (version 5.01, San Diego, CA, USA) using two-way repeated measures ANOVA. When significant interactions between experimental factors were demonstrated, *post hoc* tests were made at each time point with Tukey’s test. Differences were considered statistically significant at the *p* < 0.05 level.

## 3. Results

### 3.1. Time-Course Changes in Body Weight during the Feeding Period with Different Diets

Figure 1 shows the increment of body weight during feeding period. From the first week to the end of experiment, increment of body weight in the rats fed beef tallow and safflower oil diets were significantly higher than those in the rats fed fish oil and normal rodent diets (the 1st week, F _[3, 20]_ = 49.88, *p* < 0.01; the 2nd week, F_[3, 20]_ = 11.86, *p* < 0.01; the 3rd week, F_[3, 20]_ = 13.66, *p* < 0.01; and the 4th week, F_[3, 20]_ = 16.15, *p* < 0.01). There was no significant difference in body weight gain either between fish oil and normal rodent groups or between safflower oil and beef tallow groups during the test periods.

### 3.2. Differences in Food and Water Intake in Response to Different Diet Feeding

Daily food intake of each week in rats fed different diets was presented in Figure 2A. The rats on normal rodent diet consumed significantly more food than the rats on high-fat diets (beef tallow, safflower oil or fish oil) during the 4-week feeding period (*p* < 0.01). No significant difference in 24 h food intake was found among the 3 high-fat diet groups.

As shown in Figure 2B, the pattern of daily water intake of rats in each diet group was similar to that of daily food intake. The rats fed normal rodent diet drank significantly more water than the rats in high-fat diet groups (*p* < 0.01). There was no significant difference in 24 h water intake between beef tallow and safflower oil high-fat diet groups during the testing period. Interestingly, the rats fed fish oil diet consumed significantly less water than the rats fed safflower oil diet during the 2nd, 3rd and 4th week of feeding (*p* < 0.05).

### 3.3. Effects of Different Diets on Body Temperature

Figure 3 shows the changes in fasting body temperature in the rats of each diet group. The body temperature of the rats fed the safflower oil diet significantly increased after 4 weeks feeding (*p* < 0.01), compared to the same rats before safflower oil feeding. The body temperature in rats fed the safflower oil was also significantly higher than that in the rats of the other 3 groups at Week-4 (*p* < 0.01). No significant difference in rat body temperature was found among the normal rodent, fish oil and beef tallow groups after 4 weeks feeding.

### 3.4. Effects of Different Diets on Fat Pad Weight

The weights of fat pads (% of body weight) in epididymal, retroperitoneal, mesenteric and subcutaneous fat pads of rats fed safflower oil and beef tallow diets were significantly greater than those of rats in both normal rodent diet and fish oil groups (*p* < 0.05) (Figure 4). Interestingly, in retroperitoneal and subcutaneous fat pads of rats on safflower oil diet, the percent fat pad weight was even greater than those of rats on beef tallow diet (*p* < 0.05). The percent fat pad weight of mesenteric fat pads of rats fed the fish oil diet was less than that of rats fed the normal rodent diet (*p* < 0.05). No difference was found in percent fat pad weight of epididymal, retroperitoneal and subcutaneous fat pads of rats fed fish oil, compared with those of rats fed the normal rodent diet.

### 3.5. Effects of Different Diets on Adipose Cell Number and Cell Size

Figure 5 shows the adipocyte cell number and cell size, as counted by a Coulter electronic counter, of each fat pad in rats fed different diets. Among all groups, there were no differences in adipocyte number of epididymal and mesenteric fat pads. Adipocyte numbers in retroperitoneal fat pads of rats on safflower oil diet were significantly higher than those of rats fed with the normal rodent diet and fish oil diets (*p* < 0.05) (Figure 5A). In subcutaneous fat pads, adipocyte numbers in rats on the safflower oil diet were also significantly higher than those in rats of the other 3 groups (*p* < 0.05). No significant differences in adipocyte number in all fat pads were found between the fish oil and normal rodent diet groups.

On the other hand, adipocyte size in all fat pads of the rats fed with safflower oil and beef tallow diets was significantly greater than that of rats in the normal rodent diet and fish oil groups (*p* < 0.05) (Figure 5B). Adipocyte size in epididymal, mesenteric and subcutaneous fat pads in rats fed the safflower oil diet was also greater than that of rats fed the beef tallow diet (*p* < 0.05). In rats fed the fish oil diet, adipocyte size in the mesenteric fat pad was less, but that in subcutaneous fat pad was greater, compared with those of rats fed the normal rodent diet (*p* < 0.05). No differences were found in adipocyte sizes in epididymal and retroperitoneal fat pads between the fish oil and normal rodent diet groups, and in the retroperitoneal fat pad between the safflower oil and beef tallow diet groups.

### 3.6. Effects of Different Diets on Cholesterol Content in Fat Pads

Cholesterol content in the fat tissue of safflower oil-fed rats was significantly higher than that in beef tallow, fish oil and normal rodent diet-fed rats for all four fat depots examined (*p* < 0.05) (Figure 6A). There was no significant difference in cholesterol content for all four fat pads among beef tallow, fish oil and normal rodent diet-fed rats.

### 3.7. Effects of Different Diets on Triacylglycerol Content in Fat Pads

Tissue triacylglycerol content of retroperitoneal and mesenteric fat pads in rats fed safflower oil and beef tallow was significantly higher than that in rats fed the normal rodent diet and fish oil (*p* < 0.05) (Figure 6B). In retroperitoneal and mesenteric fat pads, there was no significant difference in rats either between beef tallow and safflower oil groups, or between fish oil and normal rodent diet groups. No significant difference in triacylglycerol content was found in epididymal and subcutaneous fat pads among all diet groups (Figure 6B).

### 3.8. Effects of Different Diets on Cholesterol Content in the Adipocytes of Different Fat Pads

Cholesterol content in epididymal fat pad adipocytes in safflower oil rats was 2.5-fold higher than beef tallow, fish oil and normal rodent diet rats (*p* < 0.05) (Figure 7A). Safflower oil-fed rats also accumulated more cholesterol in retroperitoneal fat pad adipocytes than beef tallow-fed rats (*p* < 0.05), and in mesenteric fat pad adipocytes than fish oil and normal rodent diet rats (*p* < 0.05). No significant difference in cholesterol content of subcutaneous fat pad adipocytes was found among all diet groups (Figure 7A).

### 3.9. Effects of Different Diets on Triacylglycerol Content in Adipocytes of Different Fat Pads

Triacylglycerol content in visceral fat pad adipocytes, including epididymal, retroperitoneal and mesenteric fat pads, in rats fed safflower oil and beef tallow was significantly higher than that of rats fed fish oil and normal rodent diet (*p* < 0.05). There was no significant difference in adipocyte triacylglycerol content of subcutaneous fat pads among all diet groups (Figure 7B).

## 4. Discussion

The results of our present study have provided further evidence that the type of lipids fed in high-fat diets can affect lipid accretion and adipocyte cellularity. We demonstrated for the first time that 4-week safflower oil feeding caused hyperthermia and fat accumulation in visceral fat pads, especially the epididymal fat pad, with significantly more cholesterol deposition. On the other hand, compared to normal rodent diet, the fish oil diet did not cause any significant differences in body temperature, fat deposition, or triacylglycerol and cholesterol content in adipose tissues. Interestingly, the mesenteric fat pad weight and adipocyte size in rats on the fish oil diet were even significantly smaller than those in rats fed the normal rodent diet, which further supported the observations reported by Parrish et al. that fish oil prevents fat accumulation and adipocyte growth in fat tissues [20,21].

While the amount of daily food intake in the rats with different high-fat diets (10% fat) was comparable, it was significantly less than that in rats with normal rodent diet (Figure 1). Since the high-fat diets were more calorically dense than the normal rodent diet (4% fat), these data suggested that food intake was mainly determined by caloric intake, but not by the amount of food consumed. Interestingly, compared with the rats on normal rodent diet, the rats fed high-fat diets drank significantly less water, which could be due to less food they consumed. It is well-documented that the water intake is mainly influenced by the amount of food intake. This observation further supports the concept that most of the water intake is associated with peri-prandial drinking [22].

The weight gain and fat deposition (percent fat pad weight) of the safflower oil and beef tallow diet groups were significantly higher than those in the fish oil and normal rodent diet groups (Figure 1; Figure 4), suggesting that the utilization of lipid for weight gain and fat deposition were dependent on the origin of the fat. Adipocyte size in all four fat pads of rats on beef tallow and safflower oil diets was significantly greater than those of fish oil and normal rodent diet-fed rats. Through the analysis of adipocyte number in each fat pad, we found that the safflower oil diet significantly increased the cell number in both retroperitoneal and subcutaneous fat pads after 4-week feeding.

It was previously thought that adipocyte number was determined in early life and remained constant throughout adulthood in both man and rat. Hirsch et al. reported that adult rats did not lose adipocytes when starved nor acquire new adipocytes during the period of rapid weight gain induced by hypothalamic damage [23]. In a group of non-obese adult male volunteers who gained 15–25% of their body weight as the result of prolonged high caloric intake, Salans et al. reported that the induced obesity resulted from adipocyte enlargement alone, but adipocyte numbers did not increase [24]. However, this concept has changed with the development of new technology. It is believed now that once the peak capacity for lipid-storage is reached, increases in fat cell number are triggered in high-fat fed animals [25]. Johnson et al. reported that, after treatment with gold-thioglucose, mice became obese and were found to have more adipocytes than untreated controls in the retroperitoneal depot [26]. Increases in adipocyte number have also been reported to occur spontaneously with age in the male rat [27]. Importantly, the change in adipocyte cell number is permanent, because the replacement of the high-fat diet with a low-fat diet results in a reduced body weight and adipocyte cell size, but not adipocyte cell number [25,28].

It has been reported that decreases in diet-induced thermogenesis (DIT) and fat oxidation rates in animals are responsible for greater body fat accumulation [29]. DIT occurs mainly in brown adipose tissue and is regulated by the sympathetic nervous system. A decrease in sympathetic activity in BAT reduces DIT [29]. In the present study, we found that there was an increased body temperature in rats after 4 weeks of feeding with safflower oil. The observation was consistent with the previous reports by Matsuo et al. [29] and Shimomura et al. [30]. In their studies, they found that norepinephrine turnover rate was significantly higher in interscapular BAT of rats fed a safflower oil diet than that in rats fed a beef tallow diet, resulting in an increased DIT in the former. Importantly, the differences in DIT and body fat accumulation between the two dietary groups were abolished by sympathectomy. Therefore, one of the explanations for our results is that the safflower oil diet increases sympathetic activity, leading to enhanced thermogenesis. However, this DIT effect does not seem sufficient to affect fat accumulation, because no differences in triglyceride levels and fat pad weight were observed in different fat pads between those of rats on the safflower oil diet and rats on the beef tallow diet. There are two possible explanations for the difference between our data and the observation reported by Matsuo et al. [29]; one is the amount of lipids in the diets (we used 10% and they used 20%), and the other is the period of feeding (we fed rats for 4 weeks and they fed for 8 weeks).

The rats on the safflower oil diet showed a significantly increased cholesterol content in all four fat pads, compared to that in the rats on beef tallow, fish oil or normal rodent diets (Figure 6). Vegetable polyunsaturated oil is known to decrease serum cholesterol level. In vitro studies using human adipocyte culture have shown that LDL and HDL cholesterol could be transferred to adipocytes [7,31]. Zsigmond et al. reported that safflower oil feeding resulted in higher binding of HDL2 to epididymal adipocyte membranes, which transported more esterified cholesterol to peripheral tissues than with beef tallow feeding [8,32]. Khuu et al. also reported that the fatty acid and phospholipid composition of fat tissue was affected by fatty acid and phospholipid content of food [32], and the alterations in fatty acid and phospholipid composition of adipocyte membrane changed the activity of membrane enzymes, which could also impact cholesterol uptake and accumulation [33]. These reports, including our present study, support the notion that one of the mechanisms of the cholesterol-lowering effect of safflower oil could be through cholesterol accumulation after the transfer of cholesterol from serum to peripheral fat tissue.

Analysis of adipose tissue cholesterol content revealed that safflower oil feeding resulted in accumulation of more cholesterol in abdominal, but not subcutaneous, adipose tissues (Figure 7). This result suggests that the effect of dietary fatty acid on HDL2 binding and on fatty acid composition of fat cell membrane might be different based on the location of fat pads. In fact, the lipolysis response to sympathetic nerve stimulation is also different based on fat pad location, but not based on adipocyte size [34]. Further study will be needed to confirm this possibility.

On the other hand, triacylglycerol content in adipocytes of epididymal, retroperitoneal and mesenteric fat pads in rats fed safflower oil and beef tallow was significantly higher than that in rats fed fish oil and normal rodent diets. This result was consistent with the data of fat cell size (Figure 5), because the main determinant of adipocyte size was accumulated triacylglycerol. This result also suggests that the mechanism of the serum triacylglycerol-lowering effect of a safflower oil diet could differ from that of a fish oil diet. The safflower oil diet lowered plasma triacylglycerol by the transfer and accumulation of triacylglycerol to fat tissues, but the fish oil diet did not induce the accumulation of triacylglycerol in the fat tissue; instead it reduced triacylglycerol content, compared to the safflower oil and beef tallow diets, and even the normal rodent diet in mesenteric fat pad. This effect of fish oil may be related to the fact that fish oil fatty acids, especially docosahexaenoic acid and eicosapentaenoic acid, significantly inhibit the secretion of very low density lipoprotein by the liver [35].

## 5. Conclusions

The present study demonstrates that in rats (1) feeding of safflower oil induced hyperthermia and also increases fat and cholesterol content of abdominal fat tissue, especially in epididymal fat pad; (2) feeding of beef tallow only resulted in increased fat deposition, without affecting body temperature or cholesterol content of the fat tissue; (3) feeding of fish oil showed less fat and cholesterol deposition in the adipose tissue, as compared with the safflower oil and beef tallow diets, and these results were comparable to the data observed in the rats fed the regular rodent diet. These results clearly demonstrate that the dietary lipid composition determines differences in fat deposition, adipocyte number and size, and cholesterol and triacylglycerol accumulation in adipose tissue.

## Figures and Tables

**Figure 1 nutrients-12-03241-f001:**
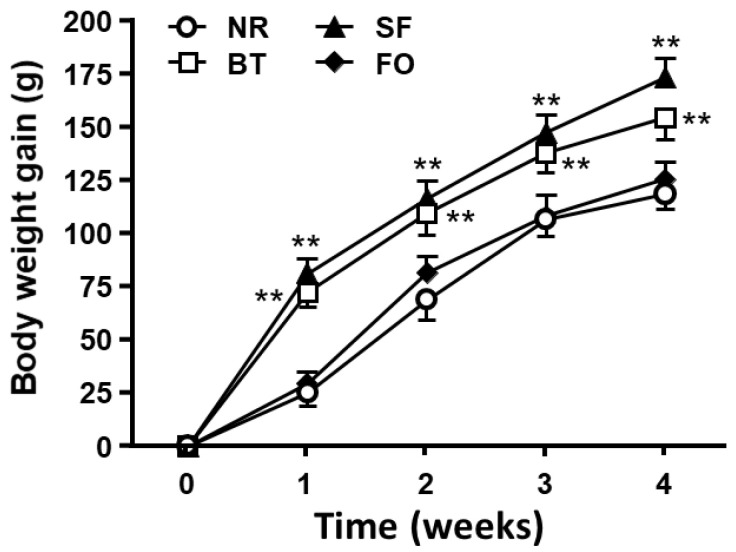
The increase of body weight in rats during 4-week feeding period with either normal rodent diet (NR, 4% fat) or 10% high-fat diets, including safflower oil (SF), beef tallow (BT), or fish oil (FO), respectively. Values are expressed as mean ± SE.; *n* = 6, *** p* < 0.01 vs. FO and NR.

**Figure 2 nutrients-12-03241-f002:**
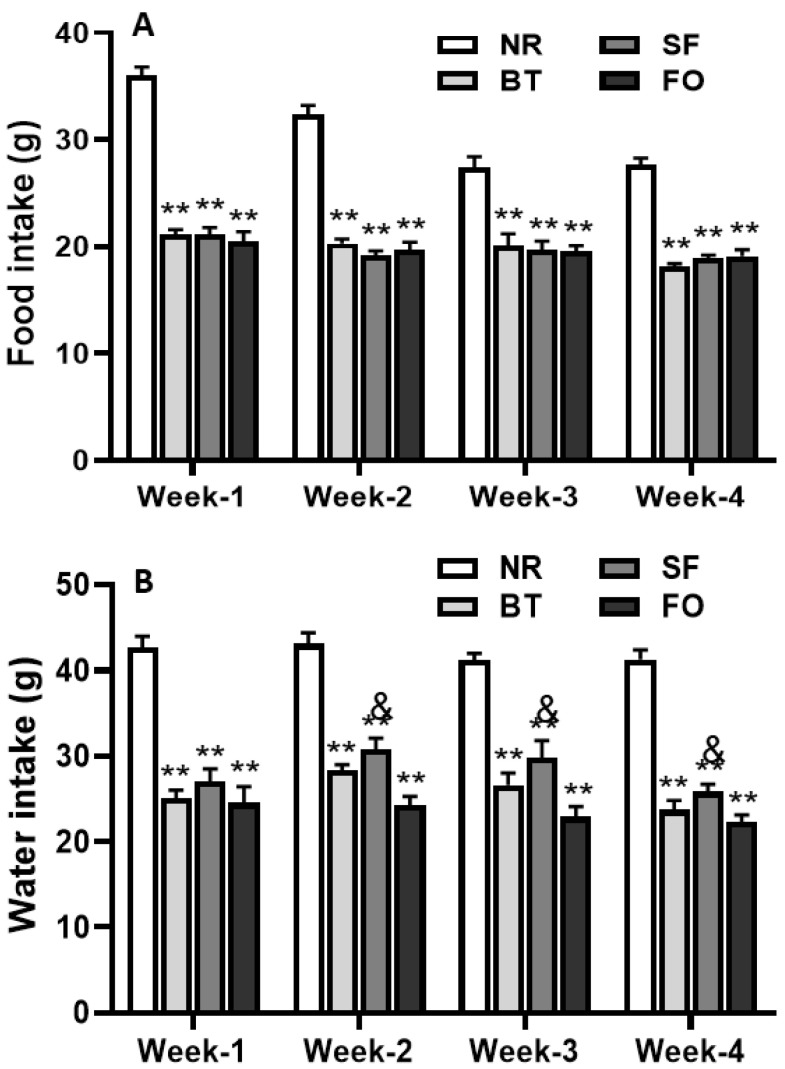
Daily food (**A**) and water (**B**) intake of each week in rats during 4-week feeding period with either normal rodent diet (NR, 4% fat) or 10% high-fat diets including beef tallow (BT), safflower oil (SF), or fish oil (FO), respectively. Values are expressed as mean ± SE.; *n* = 6, *** p* < 0.01 vs. NC; ^&^
*p* < 0.05 vs. FO.

**Figure 3 nutrients-12-03241-f003:**
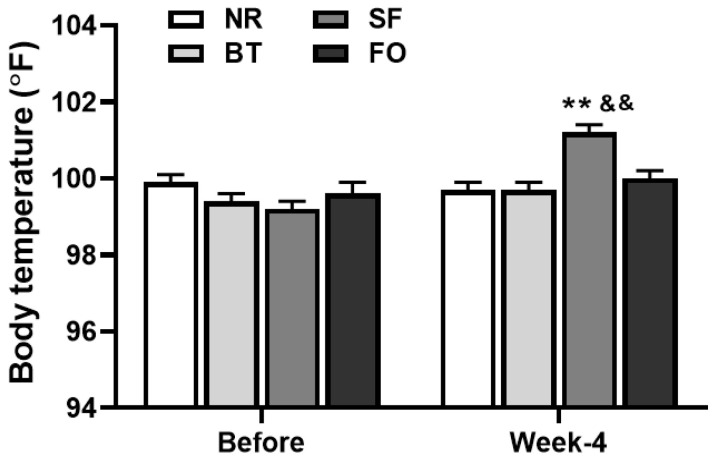
Comparison of body (rectal) temperature of rats before and after different diet feeding for 4 weeks. Values are expressed as mean ± SE.; *n* = 6, *** p* < 0.01 vs. safflower oil (SF) rats before the diet feeding, and ^&&^
*p* < 0.01 vs. the rats in other 3 groups at the same time point.

**Figure 4 nutrients-12-03241-f004:**
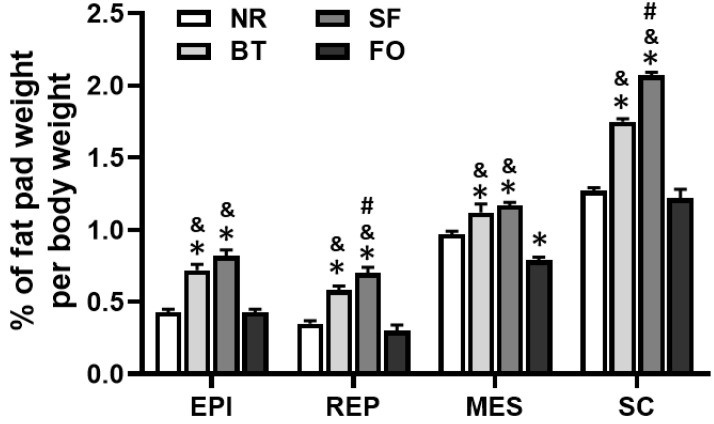
Comparison of each fat pad weight expressed as a percentage of body weight in rats after different diet feeding for 4 weeks. EPI: epididymal fat pad; REP: retroperitoneal fat pad, MES: mesenteric fat pad; and SC: subcutaneous fat pad. Values are expressed as mean ± SE.; *n* = 6, ** p* < 0.05, vs. normal rodent diet (NC); ^&^
*p* < 0.05, vs. fish oil (FO); and ^#^
*p* < 0.05, vs. beef tallow (BT).

**Figure 5 nutrients-12-03241-f005:**
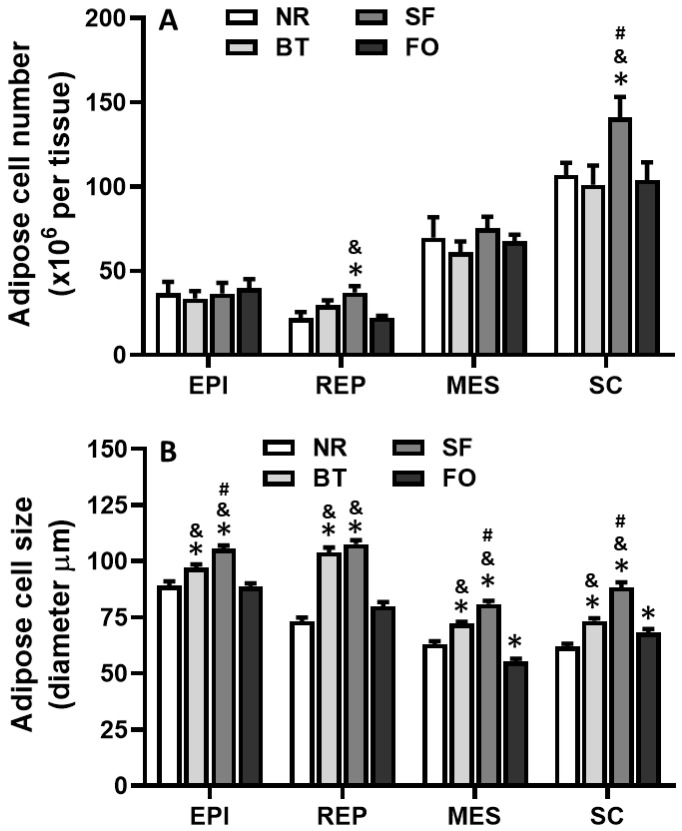
Comparison of adipose cell number (**A**) and cell size (**B**) in the epididymal (EPI), retroperitoneal (REP), mesenteric (MES) and subcutaneous (SC) fat pad of rats after feeding of different diets for 4 weeks. mean ± SE.; *n* = 6, ** p* < 0.05, vs. normal rodent diet (NC); ^&^
*p* < 0.05, vs. fish oil (FO), and ^#^
*p* < 0.05, vs. beef tallow (BT).

**Figure 6 nutrients-12-03241-f006:**
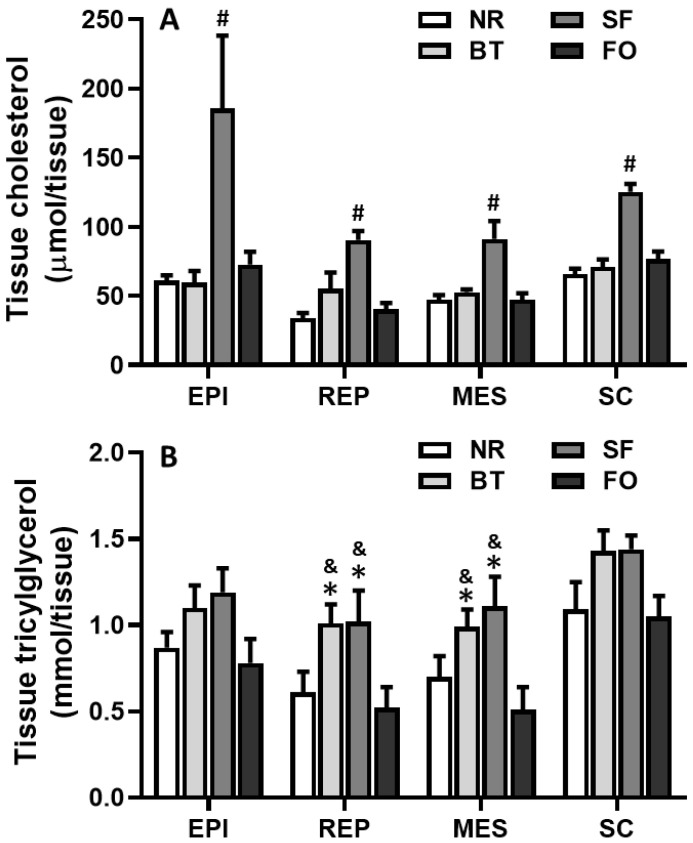
Comparison of cholesterol (**A**) and triacylglycerol (**B**) contents in the epididymal (EPI), retroperitoneal (REP), mesenteric (MES) and subcutaneous (SC) fat pads of rats after different diet feeding for 4 weeks. Values are expressed as mean ± SE.; *n* = 6, *^#^ p* < 0.05, vs. the rest 3 groups; * *p* < 0.05, vs. normal rodent diet (NC); and ^&^
*p* < 0.05, vs. fish oil (FO).

**Figure 7 nutrients-12-03241-f007:**
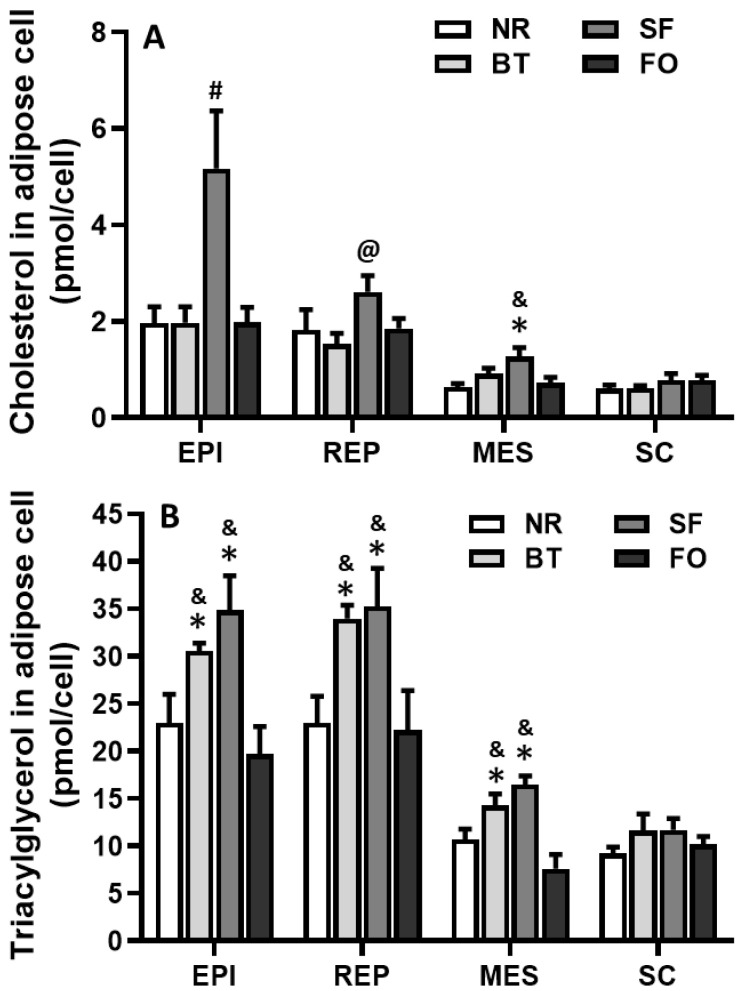
Comparison of cholesterol (**A**) and triacylglycerol (**B**) contents in the adipose cells of epididymal (EPI), retroperitoneal (REP), mesenteric (MES) and subcutaneous (SC) fat pad in rats after different diet feeding for 4 weeks. Values are expressed as mean ± SE.; *n* = 6, ^#^
*p* < 0.05, vs. the rest 3 groups; ^@^
*p* < 0.05, vs. beef tallow (BT); * *p* < 0.05, vs. normal rodent diet (NC); and ^&^
*p* < 0.05, vs. fish oil (FO).

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
