# Peer review of "Differential Effect of Four-Week Feeding of Different Dietary Fats on the Accumulation of Fat and the Cholesterol and Triglyceride Contents in the Different Fat Depots"

_nutrients, 2020, doi:10.3390/nu12113241_

Round 1

Reviewer 1 Report

The paper is still relevant to current topics. Substantially it was prepared correctly.  It is worthy of the publication in Nutrients. To further clarify the paper, the authors are encouraged to address the following points:

  1. Page 2 line 65 - Please clarify following information “During the feeding period, food and water intake, body weight and rectal temperature were measured at 1100 h”  - Especially I am interested in  1100h??? What is that value?
  2. Please describe the each type of feeding? Normal diet (4%) of fat… What type of fat was present in the normal diet ?
  3. Please clarify “AIN 76”. Does it commercial name?
  4. In my opinion, the sub-headings in the Results section sound like conclusions. I suggest replacing them as introductory sentences. Hence, the reformatting of the text in the results section is required apart from the sub-heading number 3.5
  5. Page 5 line 138 - italic font type is not required
  6. Following wording is rather incorrect „in the safflower oil and beef tallow groups. In my opinion, Authors should refer to the type of diet. Please refer this to the entire work.
  7. Page 6 line 183 “In retroperitoneal and mesenteric fat pads, there was no significant difference 183 between beef tallow and safflower oil, and between fish oil and normal rodent diet” – I am not convinced that this statement is correct, please consider it again.
  8. Please supplement the figure capture as follows: „Figure 6. Comparison of cholesterol (A) and triacylglycerol contents (B) in the epididymal (EPI), retroperitoneal (REP), mesenteric (MES) and subcutaneous (SC) fat pads of rats after different diet feeding for 4 weeks. Values are expressed as mean ± SE.; n = 6, #P < 0.05, vs. the rest 3 groups*P < 0.05, vs. normal rodent diet (NC) and &P < 0.05, vs. fish oil (FO)”
  9. Figure 7 please add “A” and “B” to this Figure - also please correct the Figure capture like Figure 6.
  10. Please pay attention to Figures: 5,6,7 to make them look similar, I mainly refer to the caption A and B (either in the drawing or outside of it)
  11. Page 9, line 273 Why did Authors write in the Conclusions section „despite the higher fat content” . Why higher fat content? What did Authors mean?
  12. In my opinion, older papers prevail in the bibliography. I propose to introduce into the Discussion section and Introduction section a few new reports from the last decade.

Reviewer 2 Report

Detailed information on the research with careful planning of work is provided. The main questions on how different fat depots are affected by chronic feeding of different fatty acids have been explained. The paper is well written with clear text and easy to read. Explanation of how beef tallow and safflower oil feeding affect the food intake and fat depot is consistent with previous literature. The main interesting outcome is how fish oil feeding decrease accumulation of fat in all fat depot similar to control with less fat intake. The results and figures were well presented and help with the understanding of the content. The conclusions are consistent with the evidence and arguments provided, addressing the main questions posed. Readers should be able to grasp the take-home message of the benefit of feeding fish oil (omega-3 fats) to decrease fat deposition in all the main fat depot in the body, but not saturated fat (beef tallow) and particular not by omega-6 fats (safflower oil).

Only minor errors are noted, that is the program used for the statistical analysis is not stated, such as whether SAS or SPSS or Excel was used. However, this is only a minor error. Another minor error is on page 10, line 281 - not sure whether the word 'please add' should be there. 

Reviewer 3 Report

The study of Liu et al., is aiming to explain the effects of chronic feeding of a high-fat  diet containing different types of lipids. They measure cholesterol and Triglyceride contents and provide some data on rectal temperature and adipocyte size and number. 

  1. Authors mention chronic feeding but the whole experimental period includes 4 weeks. I believe they should be more precise. 
  2. Introduction is poor; authors need to address more information regarding the differences among the different lipids used and they certainly need to reinforce the originality/need of  their study.
  3. “Differences were considered significant when the probability of the difference occurring by chance was less than 5 in 100 (P < 0.05).” It is widely accepted that what P value means.
  4. Methodology is poorly described regarding Tg and Chol determination, determination of size and number of cells.
  5. Moreover, count only 20 cells is not enough. Is there any other study that have validated this method?
  6. Authors need to address why the observed effect of lower water consumption in rats fed high fed diets
  7. Line 210: “Our present studies demonstrated” study instead of studies.
  8. ¿What about the numbers of the cells of adipose tissue?
  9. Authors should provide characteristic fotos of the different groups, showing the size of the cells.
  10. “Our present studies demonstrated for the first time that chronic safflower oil feeding caused 210 hyperthermia and fat accumulation in visceral fat pads” So, thermogenesis is activated? What value do authors give to this result? How is it possible to explain that higher temperature combines with higher fat depots?
  11. “One of the possible reasons is 233 that the safflower oil diet enhanced thermogenesis, and this possibility was supported by another 234 previous report demonstrating that the norepinephrine turnover in brown adipose tissue of rats fed 235 safflower oil diet was higher than in rats fed beef tallow diet [21].” It is a long shot without any protein and gene expression to support this statement.
  12. Discussion is poor but it is normal since the study is more descriptive and do not include experiments focused on the underlying mechanism that could explain the obtained results.
